# Reconstruction of Orbital Wall Fractures with a Combination of Resorbable Plates and Antibiotic-Impregnated Collagen Sheets

**DOI:** 10.3390/jcm13071900

**Published:** 2024-03-25

**Authors:** Jeeyoon Kim, Jihyoung Chang, Junho Lee, Eun-Young Rha, Jun-Hee Byeon, Jongweon Shin

**Affiliations:** 1Department of Plastic and Reconstructive Surgery, Eunpyeong St. Mary’s Hospital, College of Medicine, The Catholic University of Korea, Seoul 03312, Republic of Korea; 2Department of Plastic and Reconstructive Surgery, Uijeongbu St. Mary’s Hospital, College of Medicine, The Catholic University of Korea, Uijeongbu-Si 11765, Republic of Korea

**Keywords:** orbital wall fracture, reconstruction, antibiotic-impregnated collagen sheet, infection, postoperative antibiotics

## Abstract

**(1) Background:** Orbital wall fractures are common in maxillofacial trauma, and artificial implants are often used for reconstruction. However, there has always been concern about infection because implants are directly exposed to the airway. This study was conducted to determine the effectiveness of a combination of resorbable plates and antibiotic-impregnated collagen sheets in reconstructions of orbital fractures and to determine whether it had an effect in reducing postoperative antibiotic use. **(2) Methods:** The retrospective study was conducted on 195 patients who underwent orbital wall reconstruction from March 2019 to August 2022. The 176 patients in the control group underwent reconstruction using only resorbable plates and were administered postoperative antibiotics for 5 to 7 days. On the other hand, the 19 patients in the experimental group underwent reconstruction using a combination of resorbable plates and antibiotic-impregnated collagen sheets and only received antibiotics once before surgery. The occurrence of ocular complications, the length of hospitalization, the infection incidence rate, and the adverse effects of antibiotics were investigated. **(3) Results:** significant ocular complications were observed in the experimental group during a follow-up period of more than 1 year. Regarding postoperative infections, there were two cases of infection in the control group (infection rate: 1.14%), while no infection was found in the experimental group. The hospitalization period of the experimental group was significantly shorter than that of the control group (*p* < 0.01), and the incidence of total adverse effects of antibiotics, especially nausea, was lower in the experimental group (*p* = 0.02). **(4) Conclusions:** The combined use of resorbable plates and antibiotic-impregnated collagen sheets allows effective orbital wall reconstruction without infection, with a shorter hospital stay, and with fewer antibiotic adverse effects.

## 1. Introduction

Orbital wall fractures are a common type of fracture in maxillofacial trauma, accounting for 18.59% of such injuries [1]. The aim of orbital wall fracture reconstruction is to remove unviable bone fragments and reduce the contents of the orbit to their original position, thereby preventing complications such as disorders of the extraocular muscles or enophthalmos. To achieve this goal, various materials have been introduced to act as a barrier to the bone defect area. Autologous bone may be the most ideal [2,3], however, it has drawbacks such as donor site morbidity and an increase in surgical time. Therefore, it is common to use artificial implants, such as bioresorbable polylactide plates, nowadays [4,5].

Orbital wall fracture surgery is not a clean operation but is a clean-contaminated procedure in which the orbital cavity communicates with the airway due to the fracture [6]. Many surgeons express concerns about infections following orbital wall fracture surgeries due to direct contact between artificial materials and the airway mucosa. According to the meta-analysis study, the incidence of surgical site infections after orbital wall fracture surgery is 1.77% [7]. Although infection is rare, if an infection does occur, it can lead to orbital cellulitis, which in severe cases can result in critical outcomes such as blindness or intracranial sepsis [8,9,10]. In many cases, additional surgery is required to treat these conditions, such as the removal of the implant [11]. Consequently, surgeons have empirically used antibiotics before and after surgeries, although a consensus on the antibiotic treatment for this procedure has not yet been established [12,13].

Antibiotic-impregnated collagen sheets have been used in open wounds or to prevent wound infections such as periprosthetic infections, diabetic foot, and sternal wounds from cardiac surgery [14,15,16]. However, there have been no studies yet on its use in orbital wall fracture surgeries. In this study, a combination of commonly used resorbable plates and antibiotic-impregnated sheets was used for orbital fracture reconstruction and its effectiveness was investigated in order to effectively reduce the infection rate while eliminating surgeons’ concerns. And the effects of reducing systemic antibiotic use were also analyzed.

## 2. Methods

The study was conducted on patients who underwent surgery for orbital wall fractures between March 2019 and August 2022. Patients who underwent reconstruction using only resorbable plates were assigned to the control group, and patients who underwent reconstruction using the combination of resorbable plates and antibiotic-impregnated collagen sheets (GENTA-COLL^®^ resorb, Resorba Medical GmbH, Nuremberg, Germany) were allocated to the experimental group. This study only analyzed data from patients aged 18 to 75 years, with a unilateral fracture, and with at least a 12-month follow-up period. Patients with bilateral orbital wall fractures, signs of infection before surgery, a history of antibiotic hypersensitivity, or loss to follow-up before 12 months were excluded. Additionally, patients with a history of previous facial surgery or open fractures accompanied by external wounds were excluded.

The surgeries were performed using either a subciliary or transconjunctival approach. In both cases, pre-septal dissection was conducted to expose the arcus marginalis, followed by an incision in the periosteum and subperiosteal dissection to expose the fracture site. After debriding unvitalized bone, any reducible bone was maximally reduced, and herniated soft tissue was repositioned. Then, in the control group, only a resorbable mesh plate was inserted using the onlay graft technique for orbital restoration. In the experimental group, all processes were performed the same as in the control group, except an antibiotic-impregnated collagen sheet was attached to the bottom of the resorbable plate (mucosal side). The antibiotic-impregnated collagen sheets have a thickness of 0.5 cm. However, after about 10 s of hydration, the thickness decreases, and the sheet becomes more adherent to the implant, facilitating insertion. The implant with the antibiotic-impregnated collagen sheet attached was inserted into the bony defect area, ensuring that the sheet was positioned towards the sinus (Figure 1).

Patients in the control group received intravenous second-generation cephalosporin before the operation and the same antibiotic for 5–7 days after surgery. In contrast, patients in the experimental group received only a single perioperative antibiotic and did not receive any postoperative antibiotics. Cefotetan and flomoxef were used as second-generation cephalosporins.

The primary endpoint of the study was infection, assessed based on the CDC guidelines for organ/space surgical site infection [17]. Criteria included purulent discharge, positive culture results, abscess, and localized infection at the surgical site (characterized by erythema, swelling, heat, and tenderness) as determined by the surgeon. Surgical interventions, such as incision and drainage or implant removal, were defined as major infections, while cases managed with antibiotic treatments were only categorized as minor infections. If infection occurred, treatment was initiated with broad-spectrum antibiotics and maintained until clinical improvement was observed. If bacteria were identified in cultures, antibiotic therapy was adjusted accordingly. Decisions for surgical drainage were made based on a comprehensive evaluation of lab results and CT scans. Additionally, implant removal was performed when necessary. Other factors analyzed included duration of operation, duration of postoperative antibiotic use, length of hospital stay, and adverse effects of antibiotics. Follow-up observations were performed at 1, 4, 12, and 48 weeks after surgery, and CT scans were performed at 12 and 48 weeks.

The collected data were analyzed using the Python Scipy statistical software package, version 1.11.4. For the data of the control group and the experimental group, continuous variables were compared by means, and categorical variables were tested at expected frequencies for significant differences. The comparison of means for continuous variables began with a normality test using the Shapiro–Wilk test. A *p*-value exceeding 0.05 was considered indicative of normality. If both groups were found to follow a normal distribution, the Student’s *t*-test was used for comparison of means; if either group did not follow a normal distribution, the Mann–Whitney U test was employed. For the categorical variables, the Chi-square test was conducted if the expected frequencies in all cells were greater than five; if any cell had an expected frequency of five or less, Fisher’s exact test was performed. All tests were two-sided, and a *p*-value < 0.05 was regarded as significant.

## 3. Results

From March 2019 to August 2022, a total of 195 patients were included in the study, according to the inclusion and exclusion criteria. Among them, 176 patients who underwent reconstruction using only a resorbable plate were included in the control group, and 19 patients who underwent reconstruction using the combination of resorbable plates and antibiotic-impregnated collagen sheets were included in the experimental group.

The mean age of the control group was 48.07 ± 17.12, and that of the experimental group was 50.26 ± 18.5, with no significant difference between the two groups (*p* = 0.54). There was no difference in the gender ratio, with female to male being 1:1.67 and 1:1.71, respectively (*p* = 0.811). There was also no statistical difference in the smoking rate and diabetes prevalence in the two groups (*p* = 0.71 and *p* = 0.69, respectively), and the left and right distribution of the injured eye was also 1:1.3 and 1:1, showing no statistical significance (*p* = 0.929). Regarding the etiology of accidents, the most common mechanism of injury in both groups was a fall accident, followed by a traffic accident, assault, sports injury, and occupational injury (Figure 2). The most frequently associated facial bone fracture in both groups was the zygomatic bone, followed by the nasal bone. The follow-up period was 12.93 ± 1.02 and 12.84 ± 0.99, respectively, showing no statistical difference (*p* = 0.418). Details on patient demographics are summarized in Table 1.

There was no difference in the operation duration between the two groups. However, there was a significant difference in the period of antibiotic administration and the length of hospital stay (both *p* < 0.05).

The infection rate did not show a statistically significant difference. However, infection occurred in two cases in the control group, whereas there were no cases of infection in the experimental group. One case was a minor infection, and the other was a major infection. The minor infection occurred 7 months after surgery and was cured after 6 days of intravenous antibiotic treatment. The major infection occurred 19 months after surgery, requiring implant removal and a capsulectomy (Table 2).

Complications due to antibiotic administration in the control group included a total of 43 cases: 3 cases of skin rash, 5 cases of diarrhea, 35 cases of nausea, and no cases of kidney damage. In the experimental group, there were no adverse effects. Overall, there was a statistically significant difference in the adverse effects of antibiotics between the two groups (*p* = 0.01), with nausea showing a particularly significant difference (*p* = 0.02). During the follow-up period for the experimental group, there were no cases of ocular complications such as enophthalmos or diplopia.

### Representative Cases

**Case 1:** A 52-year-old female with no underlying disease presented to the plastic and reconstructive surgery department for a right blowout fracture caused by hitting her head against someone else’s. She reported discomfort during upper gaze but had no extraocular muscle limitation or diplopia. The preoperative CT scan revealed a fracture in the right orbital floor and herniation of the orbital soft tissue. Under general anesthesia, an open reduction was performed. Using a transconjunctival approach, the orbital floor was exposed, revealing a bony defect of 1.4 × 1.8 cm. Reconstruction was performed using an antibiotic-impregnated collagen sheet attached to a resorbable implant, slightly larger than the defect size, oriented towards the maxillary sinus. Only a single dose of flomoxef, a perioperative antibiotic, was administered. After surgery, the patient’s discomfort improved without other ocular complications, and there were no infection signs during the follow-up period. A CT scan after one year confirmed a well-reconstructed bony defect (Figure 3).

**Case 2:** An 18-year-old male presented to the emergency room for right orbital medial and floor blowout fractures that occurred after a collision during a soccer game. A physical examination revealed extraocular movement limitations on the upper gaze and associated diplopia. The preoperative CT scan showed fractures in the medial side and floor of the right orbit with soft tissue herniation. The orbital medial wall was exposed using a transcaruncular approach, and soft tissue reduction only was performed because the defect size was small. The orbital floor wall was exposed via a transconjunctival approach, revealing a bony defect of 1.8 × 2.0 cm. Orbital floor reconstruction was performed using the combination of an antibiotic-impregnated collagen sheet attached to a resorbable implant. A single dose of flomoxef was used as a perioperative antibiotic, with no further administration. The patient remained stable after surgery with no infection, and a one-year follow-up CT scan showed successful reconstruction of the orbital walls (Figure 4).

## 4. Discussion

Orbital cellulitis can result in severe complications such as blindness, meningoencephalitis, cavernous sinus thrombosis, and intracranial sepsis [8,9,10]. This infection occurs in up to 91% of cases due to infectious bacteria from the maxillary and ethmoidal paranasal sinuses invading the orbital cavity, which is typically a clean zone [10,18]. When an orbital wall fracture is present, it becomes a predisposing factor for orbital cellulitis due to the breach of the periosteum and bone, which act as barriers to the orbital cavity [19,20,21].

Materials used for orbital wall reconstruction can be broadly classified into biological materials and manufactured implants. A representative example of a biological material is an autologous bone graft, which can be considered the best reconstructive option due to its biocompatibility, solidity, vascularity, and reduced immune response [22]. However, autologous bone grafts have the disadvantage of causing donor site morbidity and prolonging surgery time. Therefore, currently, manufactured implants are mainly used in orbital wall fracture surgery.

Among manufactured implants, resorbable plates made of polylactic acid or polyglycolic acid provide temporary support until degradation and allow fibrous granulation tissue to grow into the bone defect area to reconstruct the orbital wall [23,24]. This plate is one of the most widely used implants, as it has the advantages of being easy to use, being easy to mold during surgery, and leaving no foreign substances in the body after degradation. However, one of the most significant drawbacks is the risk of infection associated with implanting a foreign material into the body. Therefore, physicians must be attentive not only to the risk of infection due to the fracture itself but also to potential infections that can arise from implants being directly exposed to the airway.

Despite some studies showing no significant difference in the incidence of infection even without using postoperative antibiotics, in many cases, antibiotics are still used after surgery [4,23]. Although there is no established regimen, prophylactic antibiotic therapy is used as a treatment to prevent such infections following reduction operations. In a survey conducted by Courtney in 2000, it was found that 82% of the 256 members of the British Association of Oral and Maxillofacial Surgeons (BAOMS) administered antibiotics for more than 5 days after surgery [25]. A survey by Brooke in 2015 showed that among 205 members of the American Society of Maxillofacial Surgeons (ASMS) and the American Association of Facial Plastic and Reconstructive Surgeons (AAFPRS), 60.4% used postoperative antibiotics for 3 to 7 days for operable facial bone fractures, and 10.2% used them for more than 7 days [13]. Each survey was observed to form the major group in their respective studies. Therefore, this study was conducted to establish a scientific basis for the use of antibiotics after orbital wall fracture surgery. Also, when integrating the findings from surveys and actual clinical practices, it is observed that postoperative antibiotics are used for approximately 5 days following orbital wall fracture surgery. Therefore, this study has been designed to simulate real clinical situations, and a control group has been established accordingly.

The reason that a definite regimen has not yet been established may be because the incidence of surgical site infection after orbital wall reconstruction is low and has not been well studied [7]. The low incidence of implant-related infections may be due to superior facial circulation compared to other body parts. Although primarily case series, there have been cases published of infection when reconstruction was performed using nonabsorbable plates such as those made of porous polyethylene (Medpor^®^) [11,26,27,28,29]. However, there have been no studies that accurately reported the infection rate when resorbable plates were used. One study of 62 patients reported two or more cases of infection [4]. Therefore, to determine the infection-prevention effect of the combination of an antibiotic-impregnated collagen sheet with a resorbable plate, we set up 176 patients who underwent reconstruction using only a resorbable plate as a control group. As a result, it was found that the infection incidence rate in the control group was 1.14% (2 cases out of 176), which was similar to the results of previous studies [4,7,30,31].

Despite the low incidence, infections do occur, and, unfortunately, they require surgical intervention in most cases [11]. Therefore, we applied antibiotic-impregnated collagen sheets as a method to effectively prevent ascending infections while reducing postoperative antibiotic use. This study marks the first use of an antibiotic-impregnated collagen sheet for orbital wall fractures. Previously, antibiotic-impregnated collagen sheets were mainly used to prevent infections in patients. In the randomized controlled trial (RCT) conducted by Friberg et al., the insertion of a gentamicin-impregnated collagen sheet into the sternal wound in high-risk patients during open cardiac surgery significantly reduced sternal wound infections in patients with diabetes or those with a body mass index of over 25 [32]. In a systematic review conducted by Nguyen et al., four studies (RCT: three, retrospective cohort study: one) were identified that researched the insertion of a gentamicin-impregnated collagen sheet into the surgical site for primary closure after excision of sacrococcygeal pilonidal sinus disease [33]. Among these, three studies confirmed that the group using the gentamicin-impregnated collagen sheet had significantly lower surgical site infections compared to the group with no insertion.

According to the Simpson et al. study, eight patients with fecal incontinence underwent implantation of a sacral nerve stimulator along with a gentamicin-impregnated collagen sheet, resulting in no infections occurring for a median duration of 89.5 days [34]. This sheet has also been used to treat implant-associated infections. According to the study by Benito-Gonzalez et al., in patients with pediatric cochlear implant-related infections that did not respond to systemic antibiotic treatment, the infection resolved after surgical debridement of the infected soft tissue followed by fixation of the implant with a gentamicin-impregnated collagen sheet [35].

The antibiotic-impregnated collagen sheet used in this study is combined with bovine collagen and gentamicin sulfate, serving as a material that locally releases gentamicin. The collagen sheet is known for its biocompatibility and absorbability, and when inserted into the body, it is known to undergo complete degradation within 4 to 8 weeks [36]. Gentamicin is an aminoglycoside antibiotic known primarily for its bactericidal effect against Gram-negative species, although it is also effective against several Gram-positive species [14]. However, the systemic use of gentamicin leading to systemic levels of gentamicin exceeding the toxicity thresholds of 10–12 mg/L can result in kidney injury and hearing loss due to inner ear injury [37]. The use of a gentamicin-containing collagen sheet maintains lower systemic gentamicin concentrations, reducing systemic toxicity while increasing the concentration of gentamicin at the locally applied site. According to a study by Leyh et al., after a local application of a gentamicin-impregnated collagen sheet in the sternal region, the mediastinal fluid concentration remained above 300 mg/L for 36 h, while the systemic level was less than 2 mg/L [38].

Although the experimental group used only perioperative antibiotics and no postoperative antibiotics, no infection occurred. This suggests that the local antibiotic effect of the collagen gentamicin sheet was effective in preventing the occurrence of ascending infections. Furthermore, this approach can reduce complications that may arise from systematic antibiotics (such as nausea, diarrhea, skin rash, etc.) and could also contribute to reducing the emergence of antibiotic-resistant bacteria.

There is also the advantage of shorter hospital stays after surgery. The average hospitalization period for the experimental group was 4.79 ± 3.91 days, and patients could be discharged immediately after postoperative pain control was completed. Since there is no need to use intravenous antibiotics, it can shorten the hospitalization period, reduce unnecessary medical costs, and accelerate the patient’s return to society. Additionally, another advantage of applying antibiotic-impregnated collagen sheets is that they help with hemostasis [15]. In a study on patients who had implanted or replaced cardiac electronic devices, there were no cases of hematoma in the group that used collagen sheets, while the incidence of hematoma was 0.7% in the group that did not use collagen sheets, which was statistically significant.

There are some limitations to this study. First, the number of subjects in the experimental group is small. Therefore, the confidence that no infection occurred in the experimental group is low. Although there was a clinical difference in the incidence of infection, no statistical difference was found (*p* = 1.0). This means that statistical significance was not achieved because the frequency of infection in the control group was also low. However, it is a meaningful result that this study is the first study to introduce the use of an antibiotic-impregnated collagen sheet in orbital wall fractures. Additionally, it is noteworthy that, despite the shorter hospitalization period and fewer adverse effects of antibiotics by reducing the use of systemic antibiotics, not a single infection occurred during the one-year follow-up period. We believe this result holds significance despite the small number of patients in the experimental group. Second, a longer follow-up period is required. The major infection in the control group occurred 19 months after surgery. Therefore, it is believed that follow-up observation over a longer period will be necessary to determine whether there is a delayed infection. Lastly, there were no cases where a collagen sheet without antibiotics was used for orbital reconstruction in this study. Since the collagen sheet takes 4 to 8 weeks to completely degrade in the body, it could also serve as a physical barrier against sinus ascending infections. Therefore, in order to further strengthen the conclusions, a study analyzing a longer follow-up period with more subjects is needed, and a study comparing a group using only collagen sheets that do not contain antibiotics is also needed.

The need for postoperative antibiotic use after orbital wall fracture surgery was mostly empirical and lacked evidence-based validity but was widely practiced due to concerns about infection. The results of this study suggest that using antibiotic-impregnated collagen sheets in combination with resorbable plates may prevent postoperative infection while reducing the use of systemic antibiotics. It also suggests that the adverse effects of systemic antibiotics can be minimized.

## Figures and Tables

**Figure 1 jcm-13-01900-f001:**
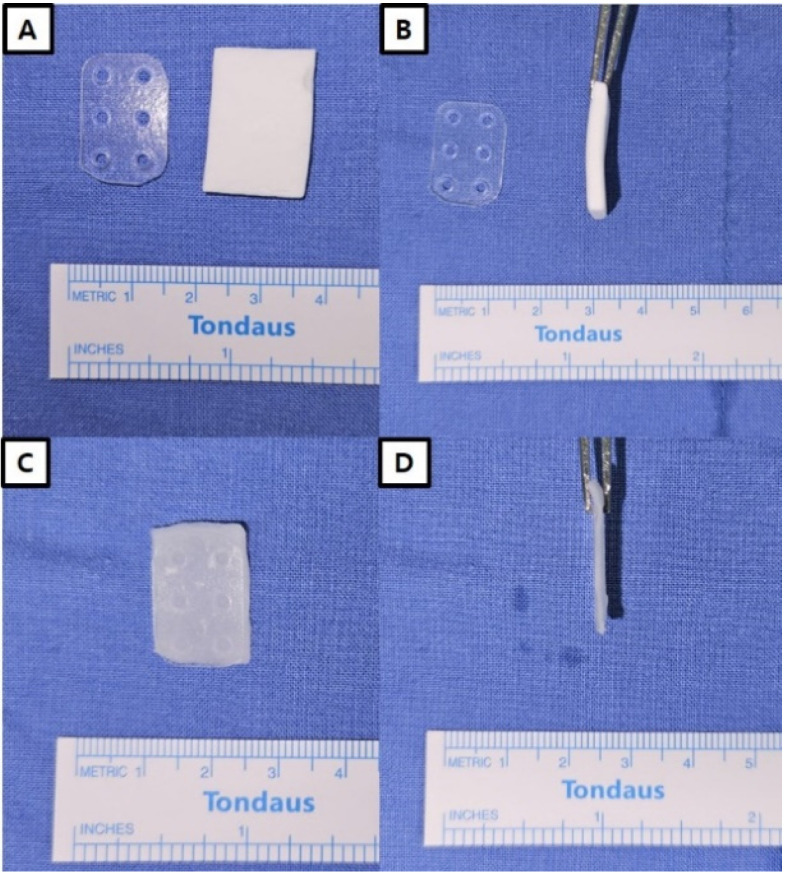
The combination of a resorbable plate and an antibiotic-impregnated collagen sheet. (**A**,**B**) The initial thickness of collagen sheet is about 0.5 cm. (**C**,**D**) After hydration, antibiotic-impregnated collagen sheet decreases in thickness and adheres to the implant, making insertion easier.

**Figure 2 jcm-13-01900-f002:**
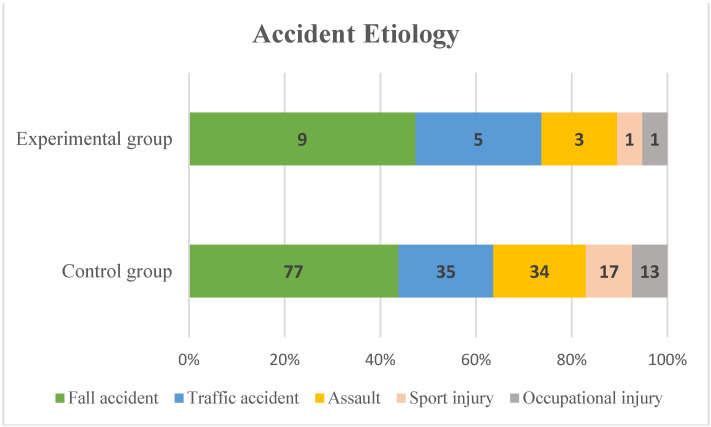
Cumulative percentage graphs of accident etiology by group.

**Figure 3 jcm-13-01900-f003:**
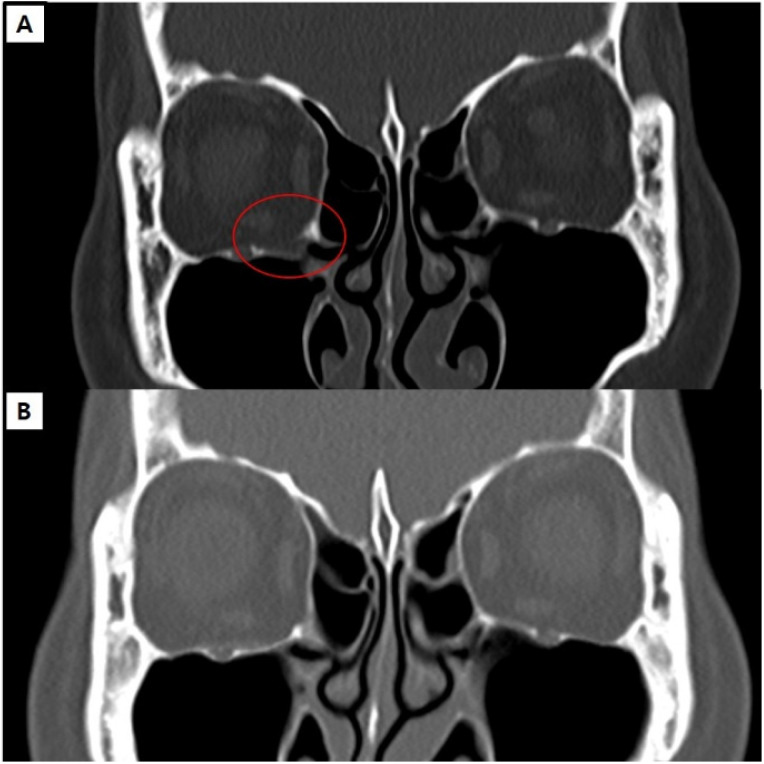
Pre- and postoperative CT scans of the case 1 patient. (**A**) Preoperative CT shows a right floor blowout fracture (red circle). (**B**) Follow-up CT scan one year after the reconstruction using the combination of a resorbable plate and an antibiotic-impregnated collagen sheet. It is observed that the bone defect was regenerated, and the reduction was well maintained.

**Figure 4 jcm-13-01900-f004:**
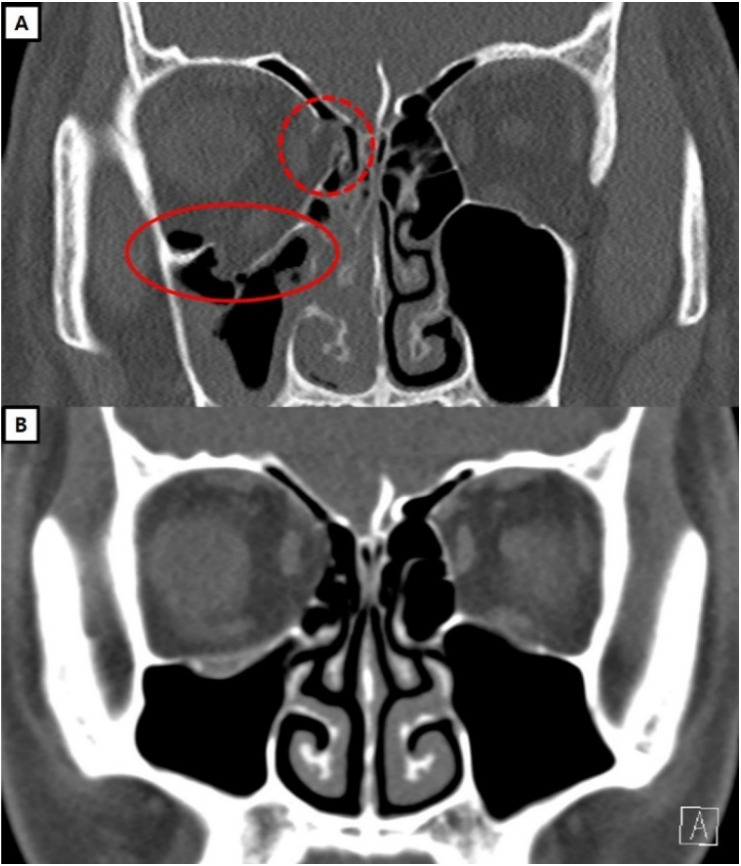
Pre- and postoperative CT scans of the case 2 patient. (**A**) Preoperative CT shows a right medial and floor blowout fracture with herniation of the orbital soft tissue (red dashed/solid circles). (**B**) The CT scan one year after surgery. It is observed that both the orbital medial wall and the floor are maintained in a well-reduced state.

**Table 1 jcm-13-01900-t001:** Patients’ demographics.

	Control Group (*n* = 176)	Experimental Group (*n* = 19)	*p*-Value
Age (mean ± SD, year)	48.07 ± 17.12	50.26 ± 18.5	0.54
Gender (F–M)	1:1.67	1:1.71	0.811
Smoking	21	3	0.71
Diabetes mellitus	16	2	0.69
Right–left distribution (right–left)	1:1.3	1:1	0.929
Accident etiology			
Fall accident	77	9	0.81
Traffic accident	35	5	0.55
Assault	34	3	1.0
Sport injury	17	1	1.0
Occupational injury	13	1	1.0
Associated facial bone fractures			
Zygoma	50	8	0.29
Nose	41	2	0.26
Follow-up period (mean ± SD, month)	12.93 ± 1.02	12.84 ± 0.99	0.48

**Table 2 jcm-13-01900-t002:** Comparative outcomes.

	Control Group (*n* = 176)	Experimental Group (*n* = 19)	*p*-Value
Operation duration (mean ± SD, min)	61.53 ± 43.7	59.15 ± 18.49	0.73
Antibiotic treatment duration (mean ± SD, day)	5.86 ± 2.35	1 ± 0	<0.01 *
Length of hospital day (mean ± SD)	6.26 ± 2.98	4.79 ± 3.91	<0.01 *
Infection	2	0	1.0
Minor infection	1	0	1.0
Major infection	1	0	1.0
Adverse effect of antibiotic use	43	0	0.01 *
Skin rash	3	0	1.0
Diarrhea	5	0	1.0
Nausea	35	0	0.02 *
Kidney injury	0	0	1.0

* Statistically significant (*p*-value < 0.05).

## Data Availability

The data presented in this study are available on reasonable request from the corresponding author. The data are not publicly available due to privacy restrictions.

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
