# Peer review of "Reconstruction of Orbital Wall Fractures with a Combination of Resorbable Plates and Antibiotic-Impregnated Collagen Sheets"

_jcm, 2024, doi:10.3390/jcm13071900_

Round 1
Reviewer 1 Report
Comments and Suggestions for Authors
This manuscript can be accepted in its present form.
Author Response
Thank you for taking the time to read our work and for your positive evaluation of our article.
Reviewer 2 Report
Comments and Suggestions for Authors
Author Response
I appreciate the time and effort that you and the reviewer have dedicated to providing valuable feedback on my manuscript. I am grateful for these insightful comments and have incorporated changes to address most of the suggestions provided. The changes are highlighted within the manuscript and point-by-point responses are provided word file.
We have incorporated the points raised by Reviewer 2 into the manuscript, highlighted in red.

Reviewer 3 Report
Comments and Suggestions for Authors
This retrospective study by Jeeyoon Kim et al was conducted to determine the effectiveness of a combination of resorbable plates and a gentamycin-impregnated collagen sheets in reconstruction of orbital fractures and to determine whether it had an effect in reducing postoperative antibiotic use. Authors reported a reducing of the incidence of infection without the use of postoperative antibiotics.
Please respond to the following concerns:
1- Detailed specification of the collagen membrane used is not clear. What is the absorption rate of this particular membrane? Also how stable is the gentamycin antibiotic in the membrane? How long does the gentamycin remain at the site?
2- Authors report on long term numbers of infection, I am not sure how a one-time locally-applied gentamycin could influence long term incidence of infection?
3- The incidence of infection in control subjects was 2 out of 176 subject and one was minor. It appears the incidence of the infection is very low anyway to justify adding antibiotic locally.
4- Can some of the beneficiary effect be contributed to the collagen membrane alone and no to the gentamycin? Since authors did not used or compare a nonantibiotic-impregnated collagen membrane to support the resorbable orbital implant used in this study.
5- The main message in conclusion is that a gentamycin-impregnated collagen membrane reduces incidences of infection in orbital surgery; however, did not show any statistical significance in the rate of infection between control and experimental group?
6- Despite difficulties in clinical research to find equal number of subjects for experimental and control groups, the extreme imbalance of control sample size vs experimental in this research is very concerning. This will drastically reduce the power of the statistical test used in this research.
Comments on the Quality of English LanguageThe english language of this paper is adequate
Author Response
I appreciate the time and effort that you and the reviewer have dedicated to providing valuable feedback on my manuscript. I am grateful for these insightful comments and have incorporated changes to address most of the suggestions provided. The changes are highlighted within the manuscript and point-by-point responses are provided word file.
We have incorporated the points raised by Reviewer 3 into the manuscript, highlighted in green.

Reviewer 4 Report
Comments and Suggestions for Authors
As you state yourself postoperative use of antibiotics lacks evidence-based validity. Thus this evidence must be achieved - in your case by adding a second control group with single shot preop antibiotics and unimpregnated sheets. All the other limitations of your study are true and can be avoided by a multi-center approach.
When you cite other studies they must be comparable unlike no. 8 where porous polyethylene mesh was used which is a good "hideout" for bacteria in contrast to smooth plyglactic sheets.
In a large German University center for maxillofacial surgery single shot antibiotic prophylaxis has been very successfully used in combination with polglactic acid sheets for the treatment of orbital floor and zygomatic fractures for decades.
Author Response
I appreciate the time and effort that you and the reviewer have dedicated to providing valuable feedback on my manuscript. I am grateful for these insightful comments and have incorporated changes to address most of the suggestions provided. The changes are highlighted within the manuscript and point-by-point responses are provided word file.

Round 2
Reviewer 2 Report
Comments and Suggestions for Authors
The authors appropriately addressed my critique and revised the manuscript accordingly..
Author Response

(The authors gave the same response as above.)

Reviewer 3 Report
Comments and Suggestions for Authors
The revised version of the manuscript has addressed most of my concerns. In light of weak statistics and small sample size , I suggest to water down the last paragraph of the discussion to teh following:
"The results of this study suggest that using antibiotic-impregnated collagen sheets in combination with resorbable plates may prevent postoperative infection, while reducing the use of systemic antibiotics"
Comments on the Quality of English LanguageThe language is ok , just need minor editorial correction by the Journal professional editors.
Author Response
I appreciate the time and effort that you and the reviewer have dedicated to providing valuable feedback on my manuscript. I changed the last paragraph of the discussion as your suggestion (highlighted in green). Also I will have the English proofreading done through an English editing company affiliated with our institution. However, the deadline is very tight, so I will consult with the editor about this issue.
Reviewer 4 Report
Comments and Suggestions for Authors
Thank you for answering some of the reviewers' qustions. There are still some critical points remaining:
1. The use of antibiotics should not be advocated to appease the anxiety of surgeons but be evidence based only.
2. The study design is not correct since there is a very small study group (n=19) vs. a large control group (n= 176) which receives a systemic antibiotic therapy . This may lead to over optimistic interpretations of the results.
The study design should be altered to a study group with impregnated sheets and a comparable control group with a preoperative one shot antibiotic prophylaxis thus already aviuding the high number (24%) of adverse side effects of a longstanding antibiotic medication .
Author Response
I appreciate the time and effort that you and the reviewer have dedicated to providing valuable feedback on my manuscript. I am grateful for these insightful comments and have incorporated changes to address most of the suggestions provided. The changes are highlighted within the manuscript and point-by-point responses are provided below.
We have incorporated the points raised by Reviewer 4 into the manuscript, highlighted in purple.
